# Accuracy of Machine Learning Algorithms for the Classification of Molecular Features of Gliomas on MRI: A Systematic Literature Review and Meta-Analysis

**DOI:** 10.3390/cancers13112606

**Published:** 2021-05-26

**Authors:** Evi J. van Kempen, Max Post, Manoj Mannil, Benno Kusters, Mark ter Laan, Frederick J. A. Meijer, Dylan J. H. A. Henssen

**Affiliations:** 1Department of Medical Imaging, Radboud University Medical Center, Radboud University, 6500HB Nijmegen, The Netherlands; evi.vankempen@radboudumc.nl (E.J.v.K.); max.post@radboudumc.nl (M.P.); anton.meijer@radboudumc.nl (F.J.A.M.); 2Clinic of Radiology, University Hospital Münster, WWU University of Münster, 48149 Münster, Germany; manoj.mannil@ukmuenster.de; 3Department of Pathology, Radboud University Medical Center, Radboud University, 6500HB Nijmegen, The Netherlands; benno.kusters@radboudumc.nl; 4Department of Neurosurgery, Radboud University Medical Center, Radboud University, 6500HB Nijmegen, The Netherlands; mark.terlaan@radboudumc.nl

**Keywords:** glioma, non-invasive molecular classification, machine learning algorithms, meta-analysis

## Abstract

**Simple Summary:**

Glioma prognosis and treatment are based on histopathological characteristics and molecular profile. Following the World Health Organization (WHO) guidelines (2016), the most important molecular diagnostic markers include IDH1/2-genotype and 1p/19q codeletion status, although more recent publications also include ARTX genotype and TERT- and MGMT promoter methylation. Machine learning algorithms (MLAs), however, were described to successfully determine these molecular characteristics non-invasively by using magnetic resonance imaging (MRI) data. The aim of this review and meta-analysis was to define the diagnostic accuracy of MLAs with regard to these different molecular markers. We found high accuracies of MLAs to predict each individual molecular marker, with IDH1/2-genotype being the most investigated and the most accurate. Radiogenomics could therefore be a promising tool for discriminating genetically determined gliomas in a non-invasive fashion. Although encouraging results are presented here, large-scale, prospective trials with external validation groups are warranted.

**Abstract:**

Treatment planning and prognosis in glioma treatment are based on the classification into low- and high-grade oligodendroglioma or astrocytoma, which is mainly based on molecular characteristics (IDH1/2- and 1p/19q codeletion status). It would be of great value if this classification could be made reliably before surgery, without biopsy. Machine learning algorithms (MLAs) could play a role in achieving this by enabling glioma characterization on magnetic resonance imaging (MRI) data without invasive tissue sampling. The aim of this study is to provide a performance evaluation and meta-analysis of various MLAs for glioma characterization. Systematic literature search and meta-analysis were performed on the aggregated data, after which subgroup analyses for several target conditions were conducted. This study is registered with PROSPERO, CRD42020191033. We identified 724 studies; 60 and 17 studies were eligible to be included in the systematic review and meta-analysis, respectively. Meta-analysis showed excellent accuracy for all subgroups, with the classification of 1p/19q codeletion status scoring significantly poorer than other subgroups (AUC: 0.748, *p* = 0.132). There was considerable heterogeneity among some of the included studies. Although promising results were found with regard to the ability of MLA-tools to be used for the non-invasive classification of gliomas, large-scale, prospective trials with external validation are warranted in the future.

## 1. Introduction

The most common primary brain tumor—glioma—is a rare cancer, but it is invariably fatal despite surgery, chemotherapy, and radiotherapy. While primary central nervous system tumors account for only 2% of primary tumors, they cause 7% of the years of life lost from cancer before age 70 [1,2,3]. Current glioma classification is based on the 2016 World Health Organization (WHO) guidelines, which differentiates subtypes of gliomas based on the presence or absence of isocitrate dehydrogenase (IDH) mutation and 1p/19q codeletion status. In addition to the mutation status, cytologic features and degrees of malignancy after hematoxylin and eosin (H&E) staining are also evaluated (Figure 1). Over the years, various other molecular biomarkers have been reported in the scientific literature, which led the European Association of Neuro-Oncology (EANO) to consider it necessary to update its guideline for the management of adult patients with gliomas [4].

Improved differentiation between the different subtypes of oligodendroglial tumors and astrocytic tumors based on neuroimaging would be beneficial, as this would facilitate the treatment planning, such as the extent of the resection margins and radiotherapy field [5]. Molecular characteristics of glioma have been shown to represent hallmark features that help clinicians to accurately define the nature of the neoplasm. For example, primary glioblastomas are characterized by a distinct pattern of genetic aberrations when compared with secondary glioblastomas, which develop by degeneration of pre-existing lower-grade gliomas [6]. Moreover, molecular characteristics are known to impact the effectiveness of certain treatment options and can therefore help to identify the most suitable treatment strategy for each patient individually [7,8]. Finally, the different subtypes of glioma are known to have different survival rates [9,10]. With regard to prognosis, patients suffering from a grade II glioma with an oligodendroglial origin have a 5-year survival rate of 81%, whereas those suffering from a grade II astrocytic glioma have a 5-year survival rate of 40%. When classified as WHO grade III, oligodendroglial tumors have better 5-year survival rates as compared to astrocytic tumors (43% vs. 20%, respectively). The patients suffering from glioblastoma (grade IV) have the poorest outcomes, with a 5-year survival rate of 5.5% [11]. In terms of treatment, preoperative distinguishing of oligodendroglial tumors from astrocytic tumors would be beneficial in facilitating the planning, extent of the resection, and the radiotherapy field [5]. Unfortunately, no visual features have yet been proven accurately enough to circumvent histopathological assessment after neurosurgical intervention. Application of machine learning algorithms (MLAs), however, could be helpful in the non-invasive characterization of gliomas [12].

As previously predicted, MLAs are increasingly becoming a critical component of advanced software systems in radiology [13,14]. MLAs concern medical imaging analysis carried out by (automatic) feature selection, followed by automatic classification. These processes detect complex patterns in images elusive to the eyes of neuroradiologists and make predictions that surpass human intelligence and human-level performance. In general, input data for MLAs consist of the imaging data themselves (e.g., different MRI sequences) and/or the segmentation of the regions of interest. Output data, on the other hand, are the desired parameters that should be extracted from the imaging data [13,14,15]. In general, the dataset is divided into two different sets: the training and the test set. The training set is used to train the performance of the MLA, indicating that the MLA is attempting to elucidate an often complex relationship between input data and output data. The test set is then used to test the actual performance of the data on a new dataset, indicating that the network has not yet been able to train on these data. The term “test set” is often used interchangeably with “validation” set. Nevertheless, only a small amount of MLAs are actually validated on a completely different, external dataset, which significantly hampers the further development of the integration of MLAs in daily practice [15]. 

With regard to the use of MLAs in neuro-oncology imaging, various reports on the use of MLAs, using a broad range of extracted features on magnetic resonance imaging (MRI), showed promising results with regard to the prediction of molecular markers and genetic alterations (e.g., IDH genotype, 1p/19q codeletion status, P53 mutations, MGMT promoter mutation, TERT promoter mutation, BRAF status, EGFR receptor mutations) [3,12,16,17,18,19]. However, one of the limitations of this type of research is the relatively limited amount of data in each study, which could possibly be overcome by a systematic review and meta-analysis of the aggregated study results [20]. The purpose of this study was to provide such an overview and perform a meta-analysis of the accuracy of MLAs in predicting gliomas’ genotype.

## 2. Materials and Methods

### 2.1. Guidelines and Registration 

A systematic review and meta-analysis were conducted following the Preferred Reporting Items for Systematic Reviews and Meta-Analyses (PRISMA) statement [21]. Prior to the initiation of the review, the study protocol was registered in the international open-access Prospective Register of Systematic Reviews (PROSPERO) under the number CRD42020191033.

### 2.2. Search Strategy, Inclusion Criteria, and Exclusion Criteria

Papers describing the use of MLAs for the binary classification of molecular characteristics of gliomas were reviewed. Databases searched for literature included Medline (accessed through PubMed), EMBASE, and the Cochrane Library. Searches were conducted from 1 April 2020 to 24 January 2021. Search strings are made available in the Appendix A and were included when they discussed the use of classification MLA methodologies on MR images in glioma patients. Next, papers must report results as mean accuracy and/or mean area under the receiver operator characteristics curve (AUC). Papers were excluded from this review when they discussed findings in animal-based studies or in non-human samples. In addition, MLA models needed to be at least internally validated. Letters, preprints, scientific reports, and reviews were included. After the removal of duplicates, the remaining papers were systematically screened on title and abstract by two researchers (E.v.K. and D.H.) independently. Non-consensus papers were identified and discussed by two researchers (E.v.K. and D.H.) to resolve disagreements and to reach consensus. Formal quality assessment tools are still lacking for this type of research [19], although a version of the TRIPOD statement tailored to machine learning methods has been announced [22].

Standardized tables were used to acquire the information of interest from the included articles. Data extracted from each study were (a) first author and year of publication, (b) size of the training set, (c) mean age of participants in the training set, (d) sex of participants in the training set, (e) size of the validation set, (f) whether there was an external validation, (g) study design, (h) architecture of the MLA algorithm(s), (i) target condition, (j) performance of the algorithm(s). Performance of the classification tools was expressed in accuracy, AUC, sensitivity, and specificity for both the training and the validation set. For studies performing external validation, externally validated data were displayed. Extracted data were cross-checked afterward, and discrepancies were resolved.

### 2.3. Statistical Analysis

Meta-analysis was conducted on the papers, which included the AUC ± standard deviation (SD) using a random-effects model to estimate the performance of the included MLA methodologies. For inclusion in quantitative analysis, studies must have reported a standard deviation, 95% confidence interval (CI), or standard error, along with the AUC-value. For the meta-analysis, the standard deviation was derived from the standard error or 95%-CI for studies not reporting the standard deviation [23]. If not provided, corresponding authors were contacted with the request to provide the necessary data to be included in the meta-analysis. Results of all appropriate studies were combined to meta-analyze the aggregated data. Then, meta-analyses were conducted on different subgroups of target conditions in order to estimate the accuracy of the algorithm for each condition separately. To be included in subgroup analysis, an additional criterium for the included studies was to describe a specific target condition (e.g., IDH and 1p/19q). Meta-analysis was performed with the use of IBM SPSS Statistics (IBM Corp. Released 2017. IBM SPSS Statistics for Windows, Version 25.0. Armonk, NY: IBM Corp.) and OpenMeta[Analyst] software (MetaAnalyst, Brown University EPBC [24]), which is the visual front-end for the R package (version 12.11.14) [25]. Results were displayed in forest plots. AUC-values of subgroups were compared by looking at the 95% confidence intervals and whether there was any overlap. The Higgins I2-test was used to test for heterogeneity between included studies with I2 > 75% deemed as considerable heterogeneity [23]. The Egger regression analysis was carried out to test for publication bias [26].

## 3. Results

A total of 1094 publications were initially retrieved through literature searches, and 724 remained after the removal of duplicates. After title and abstract screening, 215 full-text articles were assessed for eligibility. After a full-text assessment, 60 and 17 studies were included for the systematic review and meta-analysis, respectively (Figure 2). A total of 155 studies were excluded based on the full-text assessment for the following reasons: 66 studies described the segmentation of tumor-area instead of the classification of glioma; 23 studies did not report AUC as a performance metric; 14 studies discussed the classification of texture features instead of molecular characteristics of glioma; 14 studies had an incomplete description of the published data and/or methods; 14 studies did not describe an MLA model, but instead a number of combined features; 13 studies reported a too specific target condition (e.g., H3K27M and Ki-67); 5 studies used imaging techniques other than MR imaging; 3 studies had no internal validation-group; 3 studies included other brain tumors besides glioma.

### 3.1. Review of the Included Studies

A total of 60 studies [16,17,18,19,20,21,22,23,25,26,27,28,29,30,31,32,33,34,35,36,37,38,39,40,41,42,43,44,45,46,47,48,49,50,51,52,53,54,55,56,57,58,59,60,61,62,63,64,65,66,67,68,69,70,71,72,73,74] were included after the eligibility assessment. Table 1 shows the participant demographics and study characteristics. The performance evaluation of the MLA algorithms in terms of accuracy, AUC, sensitivity, and specificity for the validation set is displayed in Table 1, organized by target condition subgroups. For 12 of the included studies, an external out-of-sample validation was carried out [23,35,36,39,40,41,46,48,50,64,67,74], all other studies performed internal validation only.

Due to the high dimensionality and complexity of MRI data using different sequences, two critical steps were used to reduce the computational power needed to carry out such complex analyses. Feature selection was one key step in the discovery of predictors from high-throughput features. The included imaging genomics studies built their classification models by selecting a set of non-redundant features. The second step consists of applying a dimensionality reduction method. Often, unimodal methods are used for each dataset separately, thus failing to properly extract important, though subtle, interactions between various sequences. The least absolute shrinkage and selection operator (Lasso) method was discussed most often to perform dimensionality reduction [27,28,29,30,31,32,33,34,35,36,37,38,39,40,41,42,43,44,45]. With regard to feature selection, most papers discussed the use of a support vector machine-based recursive feature elimination (SVM-RFE) method [28,32,33,34,35,36,37,42,43,45,46,47,48,49,50,51,52,53,54,55,56,57,58,59,60,61,62,63,64,65,66,67]. The Lasso method was another often mentioned method to select features [27,28,29,30,31,32,33,34,35,36,37,38,39,40,41,42,43,44,45]. Other methods, which were frequently used for feature selection, concerned random forest classification algorithms [50,51,53,63,66,68,69,70,71,72,73,74], convolutional neural networks [37,46,48,56,57,65,75,76,77,78,79,80,81], and/or logistic regression models [33,57,82,83]. 

For the target condition, 22 studies focused on IDH mutation status, 6 on 1p/19q codeletion status, 11 on MGMT promoter methylation status, 3 on TP53 mutation status, 1 on the PTEN gene mutation, 2 on ATRX gene mutation status, 3 on the TERT promoter mutation, and 20 on the differentiation between different subtypes of glioma. Five of the included studies described multiple target conditions and/or multiple tested MLA models, each of those presented separately in Table 1. The included classification studies showed different references in classifying the presence or absence of the outcome of interest, such as standard-of-care diagnosis, (immuno)histopathology, and expert consensus. Twenty-two studies [17,18,19,23,25,28,32,33,37,39,43,45,49,51,56,61,63,66,69,70,71,74] focused on classifying the IDH mutation status in glioma. Mutations of the IDH genes serve as a diagnostic marker to diffuse WHO grade II and III gliomas as well as secondary glioblastomas. They are associated with a better prognosis in these gliomas [7]. One included study described two separate MLA methods, and therefore 23 methodologies could be analyzed. All studies used retrospectively collected data, and three [23,39,74] carried out external validation. Five studies did not report a validated AUC-value. Performance evaluation from the remaining 17 [19,25,28,32,37,43,45,49,51,56,61,63,66,69,70,71,74] studies in terms of the validated AUC ranged from 0.75 to 0.99. Additionally, 14 studies [18,19,27,32,33,37,43,45,51,56,61,63,69,71] presented the sensitivity and the specificity ranging from 54% to 98% and 67% to 99%, respectively.

Six studies [20,32,56,57,60,64] described the classification of the 1p/19q codeletion status in glioma patients, all using retrospectively collected data. The 1p/19q codeletion status is associated with better prognosis in patients with (oligodendro)glial tumors receiving adjuvant radio-chemotherapy [7]. One study [64] included an external validation. Validated AUC for classifying 1p/19q codeletion status ranged from 0.72 to 0.87 (*n* = 5) [32,56,57,60,64]. The range of the sensitivity and the specificity were 68% to 92% and 71% to 85%, respectively (*n* = 5) [32,56,57,60,64].

Regarding the classification of the MGMT promoter methylation status, all studies (*n* = 11) [31,35,44,51,52,53,54,55,56,65,67] used retrospectively collected data and two studies [35,67] had been externally validated. The importance of the MGMT promoter methylation status can be found in the fact that it is a predictive variable for the response of GBM to alkylating chemotherapy [7]. Seven studies [31,35,44,51,52,56,65] reported the validated AUC-value, which ranged from 0.54 to 0.90. Moreover, the sensitivity and specificity ranged from 67% to 94% and 54% to 97%, respectively (*n* = 7) [35,51,52,55,56,65,67].

Three studies [36,56,62] were included that described the classification of the TERT promoter mutation status in glioma. All studies used retrospectively collected data, and one study [36] was externally validated. Although still under investigation, it has been suggested that TERT promoter mutations characterize gliomas that require aggressive treatment [8]. All studies reported the validated AUC (range: 0.82–0.89). Additionally, the sensitivity and specificity ranged from 71% to 77% and 86% to 91%, respectively (*n* = 3).

### 3.2. Meta-Analysis of the Included Studies

For the classification papers, 17 out of the 60 were eligible for the meta-analysis. In total, 22 MLA methodologies, described in 17 individual studies [19,22,32,34,35,36,37,49,50,56,61,62,64,65,66,73,74] were retrieved. Meta-analysis was performed separately in subgroups for different target conditions. For inclusion in this subgroup meta-analysis, studies must focus on a specific target condition (i.e., IDH genotype, 1p/19q codeletion and MGMT- and TERT promoter status).

For the subgroup meta-analysis of the classification studies with a focus on IDH mutation status, eight MLA algorithms, originating from seven studies [19,37,49,56,61,66,74], were included. Results show an overall AUC of 0.909 (95%-CI: 0.867–0.951), as seen in Figure 3. Moreover, heterogeneity between groups, measured with Higgins I2, was estimated as 90.402% (*p* < 0.001).

The forest plot shows that the performance of the MLAs to classify molecular characteristics of glioma are centered around an AUC of 0.858 with a 95%-CI ranging from 0.812–0.904.

Three studies [32,56,64] were included in the subgroup meta-analysis of the 1p/19q codeletion status. Results of this subgroup analysis are displayed in Figure 4. The overall AUC is 0.748 (95%-CI: 0.699–0.797). Heterogeneity between groups was considered moderate (Higgins I2= 50.655% (*p* = 0.132)).

Subgroup meta-analysis of MGMT promoter methylation status included three studies [35,56,65]; see Figure 5 for an overview of the results. The overall AUC of these MLA models was estimated as 0.866 (95%-CI: 0.812–0.921). Heterogeneity between the included studies was considered very low (Higgins I2= 0% (*p* = 0.453)).

Three studies [36,56,62] were included in the subgroup analysis of the TERT promoter mutation status. Figure 6 displays the results of the meta-analysis with an estimated overall AUC of 0.842 (95%-CI: 0.783–0.901) and considered a low I2 heterogeneity of 0% (*p* = 0.582).

Classification of the 1p/19q codeletion status showed to have a significantly poorer AUC when compared to other subgroup classifications, except for the TERT mutation status classification. No significant differences in performance between the other three subgroups (i.e., IDH, MGMT, TERT) were observed due to overlap of the 95% confidence intervals.

### 3.3. Testing for Publication Bias

Egger’s regression test showed no significant publication bias with regard to MLAs to predict the molecular status of gliomas (*p* = 0.235).

## 4. Discussion

In this study, a number of studies that describe the classification of gliomas with the use of MLAs were reviewed and meta-analyzed. The overall performance of the classification tools as reported in AUC-values showed to be excellent. Subgroup analysis showed that the classification of 1p/19q codeletion status was significantly poorer than the classification performance of other molecular markers (i.e., IDH and MGMT). The observed heterogeneity between the included studies in the IDH-subgroup was considerably high.

The binary classification of various molecular characteristics of glioma with the use of artificial intelligence showed promising results with regard to future implementation in clinical practice. This implementation will have a significant impact on the care of glioma patients, as it could help to stratify patients for treatment options prior to undergoing surgery. However, clinically relevant studies need to be undertaken to increase the impact of these techniques in daily practice. For example, predicting 1p/19q codeletion status is more relevant in a subset of low-grade gliomas, as it enables to non-invasively distinguish IDH-mut astrocytoma (1p/19q intact) from oligodendroglioma (1p/19q codeleted) [62]. This is clinically relevant as the median survival of patients with these glioma subtypes is significantly different and can be impacted by the extent of the neurosurgical resection [86,87]. In addition, to further verify the performance of MLA methodologies, larger-scale multi-center studies using prospective data are required. Moreover, despite the growing knowledge and use of MLA methodologies, integration in widespread clinical practice still faces some challenges [12]. One major challenge for this implementation concerns the generalizability of these systems, as they are mostly trained on small datasets lacking external validation [88]. Considering external validation as an additional inclusion criterium for a sub-analysis, we found that no meta-analysis could be performed. The twelve papers included in this review which validated their results externally investigated IDH mutation status (*n* = 2), 1p/19q codeletion status (*n* = 1), MGMT promoter methylation status (*n* = 2), PTEN gene mutation (*n* = 1), ATRX gene mutation (*n* = 1), TERT promoter mutation (*n* = 1), and various predictions with regard to WHO grading (*n* = 4), indicating that no pooled data can be acquired from these individual studies. Although the computer-aided classification of glioma holds great potential, computer-obtained diagnosis is not likely to replace histopathologic diagnosis in the near future.

### 4.1. Implementation of Computer-Aided Approaches in Future Medicine

The diagnosis of different diseases by the use of MLAs is believed to hold great potential in modern medicine. The number of retrieved papers on this narrow topic is relatively limited, especially when compared to other reviews with a broader scope with regard to the use of artificial analysis in medical imaging analysis [22]. Nevertheless, conclusions and major limitations seem to be similar across fields. We can cautiously state that the accuracy of MLAs in the non-invasive classification of glioma holds great potential and is equal to or better than the predictions of healthcare professionals. On the other hand, the lack of external validation of the obtained results was recognized as the major limitation of the current scientific literature. Additionally, poor reporting is known to be prevalent in MLA studies, which limits reliable interpretation of the reported diagnostic accuracy and thereby hampers clinical implementation. Improving reporting and publication will enable greater confidence in the results of future evaluations of these promising technologies in medicine. When such improved confidence will be achieved, prospective evaluation should be carried out in the context of an intended clinical pathway. With regard to the context discussed in this paper, MLAs could be used on the preoperative imaging data, after which the predicted outcomes can be compared to the histopathological assessment after biopsy. Such implementation will help to elucidate whether important unknown covariates were present in the retrospective studies reviewed here. Thereafter, a randomized comparison could help to reveal and quantify possible clinical implications of implementing these MLAs in daily practice.

Furthermore, as recently suggested by Bhandari et al., a greater effort is needed to start translating these findings into an interpretable format for clinical radiology [89]. In addition, as several studies focused on singular molecular biomarkers in gliomas, it must be underlined several molecular alterations in astrocytic, oligodendroglial gliomas can occur in different combinations [79]. Consequently, a growing number of (and a combination of) different molecular tests are used to provide clinically relevant tissue-based biomarkers. Furthermore, the performance of MLAs to grade glioma according to the WHO grading score remains restricted by the interobserver variability of the neuropathological examination as reviewed by Van den Bent (2010) [90]. This clinically significant interobserver variation of the histological grading of glioma limits the diagnostic performance of other diagnostic tests (e.g., MLAs) as the neuropathological assessment was considered to be the ground truth.

### 4.2. Clinical Relevance of Computer-Aided Diagnosis

Automated diagnosis from medical imaging through artificial intelligence could help to overcome the mismatch between the increasing amount of diagnostic images and the capacity of available specialists [91]. More than 100 MLAs have now CE-marked, 57 of which can be used within neuroimaging features. Only four of these MLAs have been tailored to be used on neuro-oncology practices [91] (see https://grand-challenge.org/aiforradiology/; accessed on 5 March 2021). Only one of these software packages claims to aid in tumor differentiation (The Brain Tumours Application; Hanalytics (BioMind; https://biomind.ai/; accessed on 5 March 2021). This application focuses on the differentiation of 22 types of intracranial tumors on MRI scans, including the differentiation of astrocytoma, oligodendroglioma, and glioblastoma. Therefore, no software is commercially present to distinguish different molecular subtypes of glioma. Therefore, we conclude that the use of MLA models in daily radiological practice to non-invasively predict glioma subtype remains an important topic of future research in order to improve accuracy and commence external validation [91].

### 4.3. Strengths and Limitations 

Meta-analysis of the aggregated MLA models showed high heterogeneity between included study groups. This heterogeneity could be expected, as multiple subgroups of target conditions are included in this analysis. Subgroup meta-analyses show significantly lower heterogeneity among included groups. However, for the IDH-subgroup, estimated heterogeneity still is remarkably high (I2 > 80%). Possible explanations for this heterogeneity could be the inclusion of multiple technically different MLA methodologies, multiple included MRI protocols with different sequences (e.g., T1-weighted, T2-weighted, diffusion-weighted), and the fact that there were no specific criteria set for the target population of glioma patients. The latter indicates that there is a good chance of variety between the included groups of patients. Although not supported by the results of the Egger’s regression test, the presence of publication bias is not unlikely, there being little interest for classification tools with poor performance. The analyzed MLA methodologies showed excellent accuracy in the classification of multiple molecular characteristics of glioma. However, some deficiencies in the methods of this study should be considered. The quality of included articles was not formally assessed because no sufficient assessment tool is currently available for prediction models using MLA-techniques. The previously announced MLA variant of the TRIPOD statement could possibly be the solution to this problem for future research [22]. Moreover, external validation of the MLA methodologies was conducted for only 12 of the 60 studies. As internal validation commonly overestimates the performance, external validation of the described system is highly preferred. Lastly, a large number of the studies found in the initial search had missing reports of the validated AUC-value. Therefore, plural studies needed to be excluded from this review. Moreover, out of 60 studies included for qualitative synthesis, 43 studies did not note the 95% confidence interval, standard error, and/or standard deviation, which led to a restriction in the number of studies eligible for meta-analysis. For this matter, we recommend a standardized way of reporting MLA findings. In addition, as MLAs are statistical model-fitting methods, the reliability and performance of each study are at least partially dependent on the study sample size. Therefore, a large dataset next to the available BraTS dataset would be indispensable for this field of research.

## 5. Conclusions

This systematic review and meta-analysis show good accuracy for various MLA methodologies for the classification of molecular characteristics of gliomas, which could be beneficial for treatment planning. Remarkably, various studies did not perform external validation, causing significant limitations for these study results. Quality guidelines should be used when publishing studies on MLAs, including out-of-sample external validation and standardized reporting of obtained results.

## Figures and Tables

**Figure 1 cancers-13-02606-f001:**
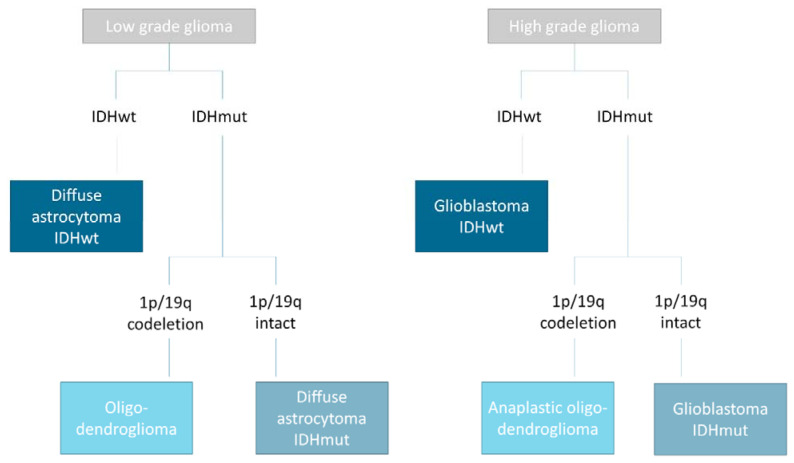
Classification of gliomas according to the WHO 2016 Classification Guidelines. IDH: Isocitrate dehydrogenase gene; mut: mutant; wt: wildtype.

**Figure 2 cancers-13-02606-f002:**
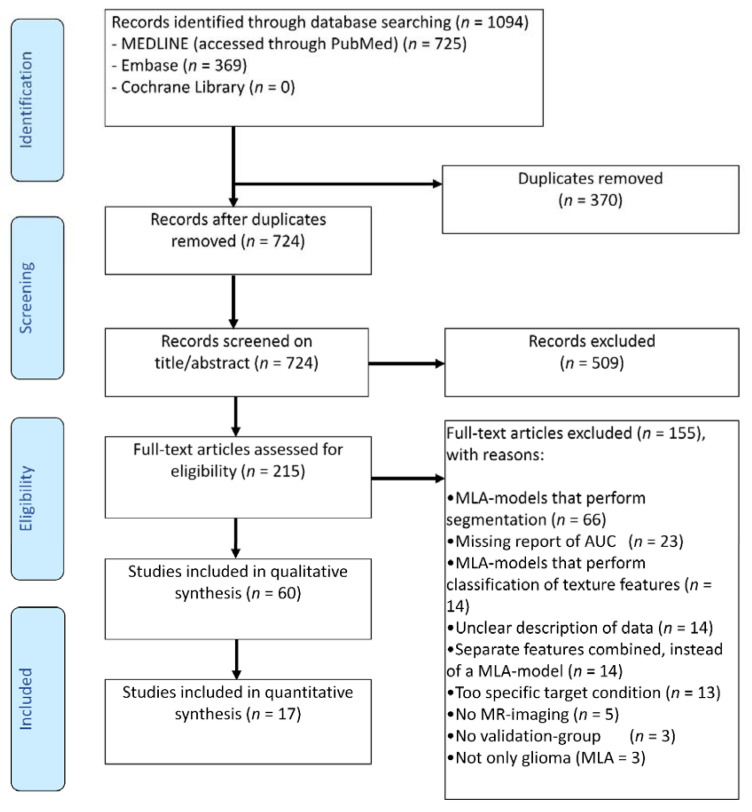
PRISMA flowchart of systematic literature search. MLA: Machine learning algorithms; MR imaging: Magnetic resonance imaging.

**Figure 3 cancers-13-02606-f003:**
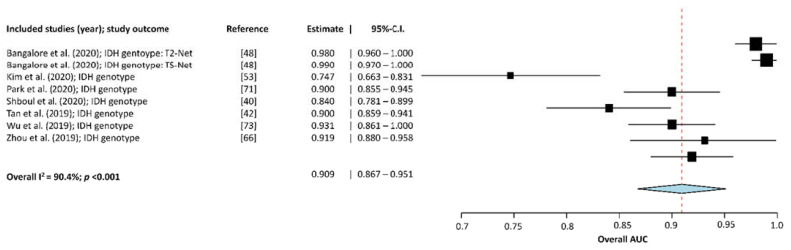
Forest-plot of the included studies that assessed the accuracy of predicting IDH mutation status in glioma. AUC, Area Under the receiver operator Curve; CI, Confidence Interval. Forest plot shows that the performance of the MLAs to classify IDH mutation status are centered around an AUC of 0.909 with a 95%-CI ranging from 0.867–0.951.

**Figure 4 cancers-13-02606-f004:**
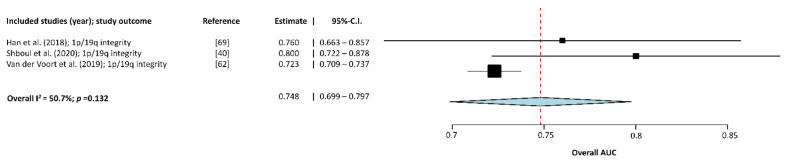
Forest-plot of the included studies that assessed the accuracy of predicting 1p/19q codeletion status in glioma. AUC, Area Under the receiver operator Curve; CI, Confidence Interval. Forest plot shows that the performance of the MLAs to classify 1p/19q codeletion status are centered around an AUC of 0.748 with a 95%-CI ranging from 0.699–0.797.

**Figure 5 cancers-13-02606-f005:**
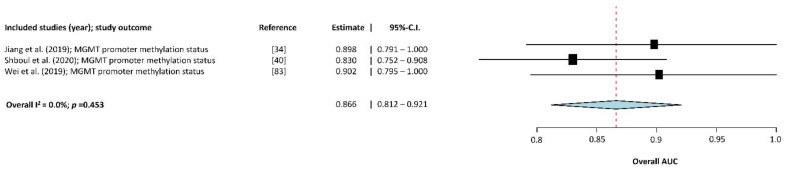
Forest-plot of the included studies that assessed the accuracy of predicting MGMT promoter methylation status in glioma. AUC, Area Under the receiver operator Curve; CI, Confidence Interval. Forest plot shows that the performance of the MLAs to classify MGMT mutation status are centered around an AUC of 0.866 with a 95%-CI ranging from 0.812–0.921.

**Figure 6 cancers-13-02606-f006:**
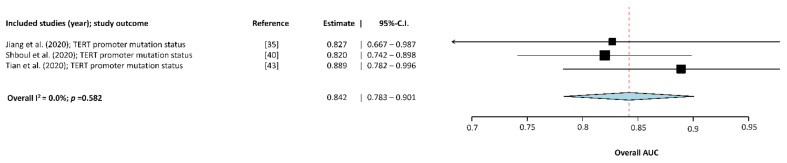
Forest-plot of the included studies that assessed the accuracy of predicting TERT promoter mutation status in glioma. AUC, Area Under the receiver operator Curve; CI, Confidence Interval. Forest plot shows that the performance of the MLAs to classify TERT promoter mutation status are centered around an AUC of 0.842 with a 95%-CI ranging from 0.783–0.901.

**Table 1 cancers-13-02606-t001:** Participant demographics, study characteristics and outcomes of the included studies.

	*Training Set*			*Validation Set*					Performance *			
First Author (Year of Publication) (Reference)	N	Age (Mean ± SD)	Gender (Male-Female)	N	Validated on an Independent External Dataset?	Input Imaging Data	MLA Method	Target Condition	Sensitivity	Specificity	AUC (±SD)	Accuracy
Ahammed Muneer (2019) [46]	389	NR	NR	168	No	T2w images; tumor segmentation	Deep CNN	Glioma grade	92.72	98.13	NR	94.64
Arita (2018) [27]	111	NR	NR	58	No	T2w-based VOI segmentation and T1w, T2w, FLAIR, and T1w +c images	Lasso and Elastic-Net Regularized Generalized Linear Model	IDH genotype	NR	NR	NR	87
Bakas (2018) [47]	86	NR	NR	NR	No	T1w, T2w, FLAIR, T1w +c images; DTI series and DSC-PWI series	Multivariate machine learning model with a Random Forests algorithm	IDH genotype	66.7	92.9	NR	88.4
Bangalore Yogananda (2020) [48]	214	NR	NR	214	No	T2w, FLAIR, and T1w +c images	3D Dense-UNet: T2-Net	IDH genotype	97	98	0.98 ± 0.146	97.14
							3D Dense-UNet: TS-Net	IDH genotype	98	97	0.99 ± 0.146	97.12
Batchala (2019) [84]	102	NR	50–52	106	No	T1w, T2w, FLAIR, and T1w +c images; DSC-PWI series	Multivariate model	1p/19q integrity	NR	NR	NR	81.1
Bonte (2016) [68]	274	NR	NR	NR	No	BraTS-data (T1w, T2w, FLAIR, and T1w +c images)	Random Forests algorithm	LGG/HGG	95.5	79.5	NR	92.3
Cao (2020) [28]	141	NR	74–67	88	No	T1w, T2w, FLAIR, and T1w +c images	Lasso and Elastic-Net Regularized Generalized Linear Model with Support vector machine classifier	LGG/HGG	NR	NR	0.915 ± 0.356	NR
Carver (2019) [29]	78	NR	NR	50	Yes	T1w, T2w, FLAIR, and T1w +c images	Lasso and Elastic-Net Regularized Generalized Linear Model	IDH genotype	NR	NR	NR	74
Chang (2018) [77]	1188	NR	NR	153	No	T1w, T2w, FLAIR, and T1w +c images	Residual CNN model	IDH genotype	NR	NR	0.93	83.0
Citak-Er (2018) [49]	43	49.5 ± 12.8	25–18	NR	No	T1w, T2w, DW images; DTI series, DSC-PWI series, and MRS	Support vector machine classifier with linear kernel and logistic regression with a Random Forests algorithm	LGG/HGG	86.7	96.4	NR	93.0
Cui (2018) [30]	40	NR	NR	NR	No	T1w, T2w, FLAIR, and T1w +c images; tumor segmentation	Lasso and Elastic-Net Regularized Generalized Linear Model	LGG/HGG	NR	NR	0.84	NR
De Looze (2018) [50]	381	NR	251–130	NR	No	Three VASARI criteria as assessed on T1w, T2w, FLAIR, and DW images	Random Forests model	IDH genotype	81	77	0.88	NR
								Glioma grade II/III	82	94	0.98	NR
								Glioma grade II/IV	100	100	1.0	NR
								Glioma grade III/IV	83	97	0.97	NR
Fan (2019) [45]	126	46.8	NR	NR	No	T1w +c images	Lasso and Elastic-Net Regularized Generalized Linear Model adopted into linear discriminant analysis and Support vector machine classifier	glioblastoma/anaplastic oligodendro-glioma	100.0	91.0	0.923	93.8
Gates (2020) [51]	23	NR	NR	NR	No	T2, ADC, CBV, and Ktrans	Random Forests algorithm	Glioma grade	NR	NR	NR	96
Han (2018) [76]	117	NR	NR	21	No	T1w, T2w, and FLAIR images	Recurrent CNN model	MGMT promoter methylation status	NR	NR	0.54	53
Han (2018) [71]	184	41.67	120–64	93	No	T2w images and T2w-based segmentation	Random Forests algorithm	1p/19q integrity	68.3	71.2	0.760 ± 0.477	70.0
Hwan-Ho (2017) [33]	108	NR	NR	NR	No	BraTS-data (T1w, T2w, FLAIR, and T1w +c images) and BraTS-segmentation	Lasso and Elastic-Net Regularized Generalized Linear Model and logistic regression	Glioma grade	88.89	90.74	0.8870	89.81
Inano (2014) [52]	33	NR	22–11	33	No	DW images, FA-maps, first eigenvalue, second eigenvalue, third eigenvalue, MD-maps, and raw T2 signal with no diffusion-weighting	Support vector machine classifiers	Glioma grade	84.8	74.5	0.912 ± 0.028	80.4
Jiang (2019) [34]	87	45.4 ± 13.1	43–44	35	Yes	T2w and T1w +c images	Lasso regression model with fusion Radiomics model and Support vector machine classifier	MGMT promoter methylation status	82.1	85.7	0.898 ± 0.323	88.6
Jiang (2020) [35]	83	45.5 ± 12.3	50–33	33	Yes	T2w and T1w +c images	Lasso regression model with radiomics signature model and Support vector machine classifier	TERT promoter mutation status	71.4	89.5	0.827 ± 0.470	84.8
Kim (2020) [53]	127	NR	68–59	28	No	T1w, T2w, FLAIR, T1w +c, DW images; DSC-PWI series	Recursive feature elimination with Support vector machine, completed with a Random Forests algorithm and a logistic regression classifier	IDH genotype	53.6	86.7	0.747 ± 0.228	NR
Kinoshita (2018) [70]	199	NR	NR	NR	No	Conventional MR sequences (NOS)	Random Forests algorithm	Glioma grade	NR	NR	0.711	64.5
Lee (2019) [54]	88	NR	47–41	35	Yes	T1w, T2w, FLAIR, DW images; DSC-PWI series	Eight machine learning classifiers: K-Nearest Neighbors, Support vector classification, Decision Tree, Random Forest, AdaBoost, Naive Bayes, Linear Discriminant Analysis, and Gradient Boosting	IDH genotype	NR	NR	NR	83.4
Li (2019) [55]	69	60.0	37–32	40	Yes	T2w and T1w +c images	Support vector machine classifier with Support vector machine classifier	PTEN genotype	86.7	70.0	0.787	82.5
Li (2018) [32]	63	43.6	25–38	91	Yes	T2w images	Lasso regression model with Support vector machine classifier	ATRX genotype	57.1	85.7	0.725	76.9
Li (2018) [33]	180	39.2	111–69	92	No	T2w images	Lasso regression model with Support vector machine classifier	P53 status	62.2	85.1	0.763	70.7
Li (2017) [56]	151	40.7 ± 10.8	81–70	151	No	T1w and FLAIR images	CNN for segmentation followed by DLR model with Support vector machine classifier	IDH genotype	94.38	86.67	0.9521	92.44
Li (2018) [77]	133	54.2	79–54	60	No	T1w, T2w, FLAIR, and T1w +c images	Multiregional Radiomics model	MGMT-methylation	NR	NR	0.88	80
Li (2018) [78]	118	53.6	70–48	107	No	T1w, T2w, FLAIR, and T1w +c images	Multiregional Radiomics models	IDH genotype	80	99	0.96	97
Liang (2018) [79]	167	52.4 ± 15.5	NR	NR	No	BraTS-data (T1w, T2w, FLAIR, and T1w +c images)	Multimodal Three-Dimensional DenseNet	IDH genotype	78.5	88.0	0.857	84.6
Lo (2020) [57]	39	NR	28–11	NR	No	T1w +c images; processed by transformed ranklet images.	Logistic regression classifier	IDH genotype	57	97	NR	90
Lu (2018) [58]	214	NR	NR	70	Yes	T1w, T2w, FLAIR, T1w +c, and DW images (T2w and DW images were optional)	Three-level machine learning model	LGG/HGG	82.5	90.5	NR	87.7
Matsui (2020) [36]	217	42	131–86	NR	No	T1w, T2w, and FLAIR images	Lasso regression model with DLR model	Grading LGG	NR	NR	NR	58.5
Mzoughi (2020) [37]	284	NR	NR	67	Yes	T1w +c images	Lasso regression model with 3D CNN model with Support vector machine classifier	Glioma grade	NR	NR	NR	96.4
Park (2020) [71]	168	NR	NR	168	No	T2w, FLAIR, and T1w +c images	Random Forests algorithm	IDH genotype	NR	NR	0.900 ± 0.298	NR
Park (2019) [72]	136	44.99 ± 12.94	65–71	99	Yes	T2w, FLAIR, and T1w +c images; DTI series	Random Forests algorithm	Glioma grade	72.6	60.4	0.72 ± 0.51	66.7
Rathore (2019) [59]	202	NR	NR	NR	No	T1w, T2w, FLAIR, and T1w +c images. Data were sometimes complemented with DTI and DSC-PWI series	CNN adjusted with a Support vector machine classifier	IDH genotype	83	86	0.85	85
								MGMT	83	85	0.84	83
Rathore (2018) [67]	111	NR	NR	NR	No	T1w, T2w, FLAIR, and T1w +c images	Support Vector Machine model with a Random Forests algorithm	MGMT-methylation	75.0	97.0	0.80	88.28
Rathore (2019) [59]	270	NR	NR	NR	No	T1w, T2w, FLAIR, and T1w +c images; DTI and DSC-PWI series	Cross-validated sequential feature selection	MGMT-methylation	NR	NR	NR	86.95
Sasaki (2018) [39]	207	NR	NR	NR	No	T1w, T2w, FLAIR, and T1w +c images	Lasso regression model with supervised component principal analysis	MGMT-methylation	NR	NR	NR	68
Sasaki (2019) [38]	201	NR	NR	NR	No	T1w, T2w, and T1w +c images	Lasso regression model with supervised component principal analysis	MGMT-methylation	67	66	NR	67
Shboul (2020) [40]	81	NR	NR	27	No	T1w, T2w, FLAIR, and T1w +c images	Lasso regression model with supervised component principal analysis and multi-resolution fractal modeling	IDH genotype	90	79	0.84 ± 0.156	NR
								1p/19q integrity	75	85	0.80 ± 0.208	NR
								MGMT-methylation	93	73	0.83 ± 0.208	NR
								ATRX genotype	69	83	0.70 ± 0.468	NR
								TERT promoter mutation status	77	86	0.82 ± 0.208	NR
Shofty (2018) [60]	47	37.7 ± 10.6	27–20	NR	No	T2w, FLAIR, and T1w +c images	Ensemble Radiomic Classifier model with a Support vector machine classifier	1p/19q integrity	92	83	0.87	87
Sun (2020) [41]	92	NR	NR	NR	No	T1w, T2w images	Lasso regression model with logistic regression models	P53 status	100	40	0.709	81.3
Takahashi (2019) [80]	44	NR	NR	11	No	DW (b1000 and b2000) images, ADC-maps, FA-maps, and MK-maps	Deep CNN model	Glioma grade	NR	NR	NR	82
Takahashi (2019) [82]	38	NR	NR	NR	No	T2w-based VOI segmentation	Logistic regression models	1p/19q integrity	69.7	73.3	0.736	71.1
Tan (2019) [42]	74	47.93 ± 13.28	45–29	31	No	FLAIR and T1w +c images; ADC-maps	Radiomics Nomogram model	IDH genotype	86.7	87.5	0.900 ± 0.116	87.1
Tian (2020) [43]	88	NR	53–35	38	No	T1w, T2w, FLAIR, and T1w +c images; MRS	Lasso regression model with Radiomics Nomogram model	TERT promoter mutation status	75.0	90.9	0.889 ± 0.335	84.2
Tongtong (2017) [61]	110	NR	NR	NR	No	3D FLAIR images	Support vector machine classifier with minimum redundancy, maximum relevance, and maximum sparse representation coefficient	IDH genotype	88	79	0.90	85
van der Voort (2019) [62]	284	NR	161–123	129	Yes	T2w and T1w +c images. Data were sometimes complemented with FLAIR images	Support vector machine classifier	1p/19q integrity	73.2	61.7	0.723 ± 0.084	69.3
Wei (2019) [83]	74	NR	42–32	31	No	FLAIR and T1w +c images; ADC-maps	Fusion Radiomics model by logistic regression modelling	MGMT promoter methylation	94.4	53.9	0.902 ± 0.305	77.4
Wu (2019) [73]	84	53.5 ± 15.0	67–59	42	No	T1w, T2w, FLAIR, and T1w +c images	Random Forests algorithm	IDH genotype	NR	NR	0.931 ± 0.233	89.5
Xi (2018) [44]	98	NR	55–43	20	Yes	T1w, T2w, and T1w +c images	Lasso regression model with Support vector machine model	MGMT promoter methylation	87.5	75.0	NR	80.0
Yang (2018) [81]	113	NR	NR	NR	No	T1w, T2w, FLAIR, and T1w +c images	CNN model	LGG/HGG	NR	NR	NR	86.7
Yu (2017) [85]	110	40.3 ± 11.3	54–56	30	No	FLAIR images	Radiomics model	IDH genotype	88	67	0.79	83
Zhang (2017) [63]	90	51.4	52–38	30	No	T1w, T2w, FLAIR, T1w +c, and DW images	Random Forests algorithm	IDH genotype	NR	NR	0.9231	89
Zhang (2018) [64]	73	NR	NR	30	No	T1w, T2w, FLAIR, and T1w +c images	Support vector machine-based recursive feature elimination	IDH genotype	85.0	70.0	0.792	80.0
								P53 status	84.6	85.7	0.869	85.0
Zhang (2020) [65]	108	NR	61–47	NR	No	DTI series	CNN model with a Support vector machine classifier	LGG/HGG	98	86	0.93	94
								Glioma grade III/IV	98	100	0.99	98
Zhao (2020) [74]	36	45.0 ± 14.4	19–17	36	No	FLAIR and T1w +c images	Random Forests algorithm	Glioma grade II/III	77.8	78.3	0.861 ± 0.240	78.1
Zhou (2019) [66]	538	NR	303–235	206	Yes	FLAIR and T1w +c images	Random Forests algorithm with a Support vector machine classifier	IDH genotype	NR	NR	0.919 ± 0.286	NR

Legend: ADC: Apparent diffusion coefficient; ARTX: Alpha thalassemia/mental retardation syndrome X linked gene; BraTS: Brain Tumor Segmentation Challenge; CNN: Convolutional neural network; DW: Diffusion-weighted images; DTI: Diffusion tensor imaging; DSC-PWI: Dynamic susceptibility contrast perfusions weighted imaging; FA: Fractional anisotropy imaging; FLAIR: fluid attenuated inversion recovery; HGG: High grade glioma; IDH: Isocitrate dehydrogenase gene; LGG: Low-grade glioma; MGMT: O6-Methylguanine-DNA Methyltransferase; MRS: Magnetic resonance spectroscopy imaging; TERT: Telomerase reverse transcriptase gene; T1w: T1-weighted images; T1w +c: T1-weighted post-contrast images; T2w: T2-weighted images; VASARI: Visually AcceSAble Rembrandt Images. * If cross-validation was used, the Performance values of the cross-validation set were provided here. When the dataset was split into training/validation/test sets, the Performance evaluation values with regard to the investigated outcome (e.g., IDH genotype) of the Validation set were provided here.

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
