# Peer review of "Accuracy of Machine Learning Algorithms for the Classification of Molecular Features of Gliomas on MRI: A Systematic Literature Review and Meta-Analysis"

_cancers, 2021, doi:10.3390/cancers13112606_

Round 1
Reviewer 1 Report
Accuracy of Machine Learning Algorithms for the Classification of Molecular Features of Gliomas on MRI: a Systematic Literature Review and Meta-Analysis
The authors presented a method treatment planning and prognosis in glioma treatment is based on classification in to low and high grade, oligodendroglioma or astrocytoma using machine learning method in MRI imaging. Meta-analysis has been done and following that authors showed adequate accuracy for all subgroups.
This paper has several strong points that I can mention:
- The subject of this research is interesting and it will make a lots of readers interested and absorbed to this manuscript;
- The manuscript is made in a good structure and authors managed to mention their points clearly.
However, there are several points that must be corrected before the manuscript goes for publication:
- A major point about this paper would concern lack of discussions on using computer aided approaches in medicine. There are many great works out there and this article reviewed about 100 references which I think this needs to be more highlighted.
- Another point can be related to more explicit description for each subsection.
- There is need to provide much better representation of this diseases with underlying causes.
- The major problem of this paper is lack of better designed graphs and tables and proper categorization of them. The current figures are outcomes of programing without any design and organization. This is absolutely important to be fixed.
- Another point can be related to not having the scientific continuous connections. Since you are using ML algorithms and deep learning, I would suggest more focus on this part. This needs to be emphasize as well due to its vital role in the process.
- Moreover, there should be some summaries of updated formulations which represents the interest of the authors to in depth analysis.
- I would suggest adding a subsection only for feature selection and dimensionality reduction in the medical imaging.
Major editorial comment:
Please make your graph more adjusted with high quality. They are readable but can be made with higher quality.
In overall, the manuscript is adequate. Therefore, I recommend major revision.
Thank you
Author Response
Author's response to Reviewer 1
Accuracy of Machine Learning Algorithms for the Classification of Molecular Features of Gliomas on MRI: a Systematic Literature Review and Meta-Analysis
The authors presented a method treatment planning and prognosis in glioma treatment is based on classification in to low and high grade, oligodendroglioma or astrocytoma using machine learning method in MRI imaging. Meta-analysis has been done and following that authors showed adequate accuracy for all subgroups. This paper has several strong points that I can mention:
The subject of this research is interesting and it will make a lots of readers interested and absorbed to this manuscript;
The manuscript is made in a good structure and authors managed to mention their points clearly.
However, there are several points that must be corrected before the manuscript goes for publication:
A major point about this paper would concern lack of discussions on using computer aided approaches in medicine. There are many great works out there and this article reviewed about 100 references which I think this needs to be more highlighted.
Thank you for your meticulous reading of our manuscript. We added additional sections to the Discussion section of the manuscript as requested by the reviewer. These sections are printed below as well.
4.1. Implementation of computer-aided approaches in future medicine
Diagnosis of different diseases by use of MLAs is believed to hold great potential in modern medicine. The number of the retrieved papers on this narrow topic is relatively limited, especially when compared to other reviews with a broader scope with regard to the use of artificial analysis in medical imaging analysis (61).61Nevertheless, conclusions and major limitations seem to be similar across fields. We can cautiously state that the accuracy of MLAs in non-invasive classification of glioma hold great potential and is equal to or better than the predictions of healthcare professionals. On the other hand, the lack of external validation of the obtained results was recognized as the major limitation of the current scientific literature. Additionally, poor reporting is known to be prevalent in MLA studies, which limits reliable interpretation of the reported diagnostic accuracy and thereby hampers clinical implementation. Improving reporting and publication will enable greater confidence in the results of future evaluations of these promising technologies in medicine. When such improved confidence will be achieved, prospective evaluation should be carried out in the context of an intended clinical pathway. With regard to the context discussed in this paper, MLAs could be used on the preoperative imaging data after which the predicted outcomes can be compared to the histopathological assessment after biopsy. Such implementation will help to elucidate whether important unknown covariates were present in the here reviewed retrospective studies. Thereafter, a randomized comparison could help to reveal and quantify possible clinical implications of implementing these MLAs in daily practice. Furthermore, as recently suggested by Bhandari et al., a greater effort is needed to start translating these findings into an interpretable format for clinical radiology [79]. In addition, as several studies focused on singular molecular biomarkers in gliomas, it must be underlined several molecular alterations in astrocytic, oligodendroglial gliomas can occur in different combinations [80]. Consequently, a growing number of (a combination of) different molecular tests are used to provide clinically relevant tissue-based biomarkers. Furthermore, the performance of MLAs to grade glioma according to the WHO grading score remains restricted by the interobserver variability of the neuropathological examination as reviewed by Van den Bent (2010) [81]. This clinically significant interobserver variation of the histological grading of glioma limits the diagnostic performance of other diagnostic tests (e.g., MLAs) as the neuropathological assessment was considered to be the ground truth. The aforementioned steps are pertinent to ensure that studies on the use of MLAs in non-invasive classification of glioma are of sufficient quality to evaluate the performance of the algorithms and will contribute to improved health care in the daily clinical practice.
4.2. Clinical relevance of computer-aided diagnosis
Automated diagnosis from medical imaging through artificial intelligence could help to overcome the mismatch between the increasing amount of diagnostic images and the capacity of available specialists (62). More than 100 MLAs have now CE-marked, 57 of which can be used within neuroimaging features. Only four of these MLAs have been tailored to be used on neuro-oncology practices (63)(see https://grand-challenge.org/aiforradiology/). Only one of these software packages claims to aid in tumor differentiation (The Brain Tumours Application; Hanalytics (BioMind; https://biomind.ai/). This application focuses on the differentiation of 22 types of intracranial tumors on MRI scans, including the differentiaton of astrocytoma, oligodendroglioma and glioblastoma. Therefore, no software is commercially present to distinguish different molecular subtypes of glioma. Therefore, we conclude that the use of MLA-models in daily radiological practice to non-invasively predict glioma subtype remains an important topic of future research in order to improve accuracy and commence external validation.
Another point can be related to more explicit description for each subsection. There is need to provide much better representation of this diseases with underlying causes.
More information on this topic was added to the Introduction section. It now reads:
“Current glioma classification is based on the 2016 World Health Organization (WHO) guidelines, which differentiates subtypes of gliomas based on the presence or absence of isocitrate dehydrogenase (IDH) mutation and 1p/19q codeletion status. In addition to the mutation status of both IDH and the 1p/19q chromosome, cytologic features and degrees of malignancy after hematoxylin and eosin (H&E) staining are evaluated (see Figure 1). Over the years, various other molecular biomarkers have been reported in the scientific literature which led the European Association of Neuro-Oncology (EANO) to consider it necessary to update its guideline for the management of adult patients with gliomas [4].
Improved differentiation between the different subtypes of oligodendroglial tumors and astrocytic tumors based on neuroimaging would be beneficial as this would facilitate the treatment planning, such as the extent of the resection margins and radiotherapy field [4].Molecular characteristics of glioma have been shown to represent hallmark features which help clinicians to accurately define the nature of the neoplasm. For example, primary glioblastomas are characterized by a distinct pattern of genetic aberrations when compared with secondary glioblastomas, which develop by degeneration of pre-existing lower grade gliomas [5]. Also, molecular characteristics are known to impact the effectivity of certain treatment options and can therefore help to identify the most suitable treatment strategy for each patient individually [6,7]. Finally, the different subtypes of glioma are known to have different survival rates [8,9]. With regard to prognosis, patients suffering from a grade II glioma with oligodendroglial origin have a 5-year survival rate of 81%, whereas those suffering from a grade II astrocytic glioma have a five-year survival rate of 40%. When classified WHO grade III, oligodendroglial tumors have better 5-year survival rates as compared to astrocytic tumors (43% vs. 20% respectively). The patients suffering from glioblastoma (grade IV) have the poorest outcomes with a 5-year survival-rate of 5.5% [10]. In terms of treatment, preoperative distinguishing of oligodendroglial tumors from astrocytic tumors would be beneficial in facilitating the planning, extent of the resection and radiotherapy field [4].”
The major problem of this paper is lack of better designed graphs and tables and proper categorization of them. The current figures are outcomes of programing without any design and organization. This is absolutely important to be fixed.
Graphs and tables were re-designed and properly organized as requested by the reviewer. Please see the revised manuscript.
Another point can be related to not having the scientific continuous connections. Since you are using ML algorithms and deep learning, I would suggest more focus on this part. This needs to be emphasize as well due to its vital role in the process.
An improved scientific continuous connection was made throughout the manuscript by revision. The topic of MLAs is now more properly introduced in the Introduction. Also, a more in depth discussion on MLAs is provided in the Discussion section. Please see the revised manuscript for more information.
Moreover, there should be some summaries of updated formulations which represents the interest of the authors to in depth analysis.
As requested, we provided more information on the clinical importance of knowing the genotypic alterations in gliomas in the revised version of this manuscript. This description of the investigated genes and their individual clinical significance are provided in the Results section:
“For the target condition, 22 studies focused on IDH mutation status, 6 on 1p/19q codeletion status, 11 on MGMT promoter Methylation status, 3 on TP53 mutation status, 1 on the PTEN gene mutation, 2 on ATRX gene mutation status, 3 on the TERT promoter mutation, and 20 on the differentiation between different subtypes of glioma. Five of the included studies described multiple target conditions and/or multiple tested MLA-models, each of those presented separately in Table 1. The included classification-studies showed different references in classifying the presence or absence of the outcome of interest, such as standard-of-care diagnosis, (immuno)histopathology and expert consensus.
Twenty-two studies [17-19,23,24,27,36,38,42,44-46,50,52,57,62,64,67,70-72,75] focused on classifying the IDH mutation status in glioma. Mutations of the IDH genes serves as a diagnostic marker for diffuse WHO grade II and III gliomas as well as secondary glioblastomas. It is associated with a better prognosis in these gliomas [80]. One included study described two separate MLA-methods, and therefore 23 methodologies could be analyzed. All studies used retrospectively collected data and three [23,38,75] carried out external validation. Five studies did not report a validated AUC-value. Performance evaluation from the remaining 17 [19,24,27,36,42,44,45,50,52,57,62,64,67,70-72,75] studies in terms of the validated AUC ranged from 0.75 to 0.99. Additionally, 14 studies [18,19,27,36,42,44-46,52,57,62,64,70,72] presented the sensitivity and the specificity ranging from 54% to 98% and 67% to 99%, respectively.
Six studies [20,31,57,58,61,65] described the classification of the 1p/19q codeletion status in glioma patients, all using retrospectively collected data. 1p/19q codeletion status is associated with better prognosis in patients with (oligodendro)glial tumors receiving adjuvant radio-chemotherapy [80]. One study [65] included an external validation. Validated AUC for classifying 1p/19q codeletion status ranged from 0.72 to 0.87 (n=5) [31,57,58,61,65]. Range of the sensitivity and the specificity were 68% to 92% and 71% to 85%, respectively (n=5) [31,57,58,61,65].
Regarding the classification of the MGMT promoter methylation status, all studies (n=11) [30,34,43,52-57,66,68] used retrospectively collected data and two studies [34,68] had been externally validated. The importance of the MGMT promoter methylation status can be found in the fact that itself is a predictive variable for response of GBM to alkylating chemotherapy. Seven studies [30,34,43,52,53,57,66] reported the validated AUC-value, which ranged from 0.54 to 0.90. Besides, the sensitivity and specificity ranged from 67% to 94% and 54% to 97%, respectively (n=7) [34,52,53,56,57,66,68].
Three studies [35,57,63] were included that described the classification of the TERT promoter mutation status in glioma. All studies used retrospectively collected data, and one study [35] was externally validated. Although still under investigation, it has been suggested that TERT promoter mutations characterize gliomas which require aggressive treatment [84]. All studies reported the validated AUC (range: 0.82-0.89). Additionally, the sensitivity and specificity ranged from 71% to 77% and 86% to 91%, respectively (n=3).”
I would suggest adding a subsection only for feature selection and dimensionality reduction in the medical imaging.
As requested, subsections on feature selection and dimensionality reduction were added to the revised manuscript. Please see the revised Results section where it reads:
“Due to the high dimensionality and complexity of MRI data using different sequences, two critical steps were used to reduce the computational power needed to carry out such complex analyses. Feature selection was one key step in the discovery of predictors from high-throughput features. The included imaging genomics studies built their classification models by selecting a set of non-redundant features. The second step consists of applying a dimensionality reduction method. Often, unimodal methods are used for each dataset separately, thus failing to properly extract important, though subtle interactions between various sequences. The least absolute shrinkage and selection operator (Lasso) method was discussed most often to perform dimensionality reduction [25-45]. With regard to feature selection, most papers discussed the use of a support vector machine-based recursive feature elimination (SVM-RFE) method [26,32-35,40,41,43-67]. The Lasso method was another often mentioned method to select features [25-45]. Other methods which were frequently used for feature selection concerned random-forest classification algortihms [50,51,53,63,66,68-74], convolutional neural networks [35,46,48,56,58,65,75-81] and/or logistic regression models [31,57,82,83].”
Major editorial comment:
Please make your graph more adjusted with high quality. They are readable but can be made with higher quality.
Graphs and tables were re-designed and properly organized as requested by the reviewer. Please see the revised manuscript.
In overall, the manuscript is adequate. Therefore, I recommend major revision.
Reviewer 2 Report
Major points
- “17 individual studies [19,22,31,33-36,50,51,57,62,63,65-67,74,75],” Prediction targets of these 17 papers are different. Therefore, Figures 3 and 8 are meaningless. In my opinion, because Figures 3 and 8 are meaningless, the meta-analysis of this paper is useless.
- “For the classification-papers, 17 out of the 60 were eligible for the meta-analysis.” How are the 17 papers selected from the 60 papers? This selection largely affected this paper.
- “However, the main bottleneck of this type of research is the relative limited amount of data, which could possibly be overcome by systematic review and meta-analysis of the aggregated study results“ The systematic review is valuable, but I speculate that the main bottleneck cannot be overcome by the systematic review.
- Ref. 10. “Therefore, as many previous review papers have pointed out, the existence of large data sets is indispensable for promoting ML study.” As shown in Ref. 10, I speculate that large dataset is more valuable than meta-analysis and/or systematic review.
- “MLA-models needed to be at least internally validated.“ In several papers of Table 1, number of Validation set is NR. Are these papers valid for this review paper? If number of Validation set is not reported, the papers might be low-quality.
- “Five of the included studies described multiple target conditions and/or multiple tested MLA-models, each of those presented separately in Table 2.” Where is Table 2?
Minor points
- “Papers describing the use of MLAs for the classification of molecular characteristics of gliomas were reviewed.” Please clarify that authors included the paper where binary classification of molecular characteristics of gliomas was evaluated using MLAs.
- “. [14]” typo.
- “For 12 of the included studies an external out-of-sample validation was carried out” Please summarize the performance in external validation as Table for the 12 studies. Please include number of external validation set in the Table.
- Table 1. If cross validation is used, how are numbers of Training set and Validation set reported?
- Table 1. If dataset is divided into training/validation/test sets, how are numbers of these sets reported?
- Abbreviation should be added to Table 1. For example, “T1w+c”, “NR”, “LGG/HGG”
- Why is 95% CI of AUC higher than 1.0 in Figures?
Author Response
Accuracy of Machine Learning Algorithms for the Classification of Molecular Features of Gliomas on MRI: a Systematic Literature Review and Meta-Analysis
Major points
“17 individual studies [19,22,31,33-36,50,51,57,62,63,65-67,74,75],” Prediction targets of these 17 papers are different. Therefore, Figures 3 and 8 are meaningless. In my opinion, because Figures 3 and 8 are meaningless, the meta-analysis of this paper is useless.
We agree with the reviewer that due to differences in prediction targets, the meta-analysis represented by Figure 3 is rather meaningless. Nevertheless, Figure 8 shows the area under the curve of papers which use MLAs in the non-invasive assessment of molecular diagnosis of gliomas. Although in this subanalysis the targeted genes are different for different papers, the Funnel plot still shows that no significant publication bias exists within this field of scientific research. For that reason, we removed Figure 3 from the revised manuscript and renumered all Figures. Nevertheless, we would like to opt to maintain the old Figure 8 for the aforementioned reasons.
“For the classification-papers, 17 out of the 60 were eligible for the meta-analysis.” How are the 17 papers selected from the 60 papers? This selection largely affected this paper.
We agree with the reviewer that the fact that we only could include 17 out of 60 papers is a drawback of this manuscript. However, by using strict inclusion and exclusion criteria with regard to the quantifiable outcomes, we could create a rather homogeneous collection of papers on the use of MLAs in predicting molecular status of glioma. We elaborated on this more in the Materials and Methods section. In addition, the increase the number of included papers, one of the investigators (D.H.) contacted the corresponding authors of the specific papers with the request to provide more detailed information. This is now better explained under the Materials and Methods section where it reads:
“Meta-analysis was conducted on the papers which included the AUC ± standard deviation (SD) using a random effects model to estimate the performance of the included MLA-methodologies. For inclusion in quantitative analysis, studies must have reported a standard deviation, 95% confidence interval (CI), or standard error, along with the AUC-value. For meta-analysis, standard deviation was derived from the standard error or 95% CI for studies not reporting the standard deviation [13]. If not provided, corresponding authors were contacted with the request to provide the necessary data to be included in the meta-analysis. Results of all appropriate studies were combined to meta-analyze the aggregated data. Then, meta-analyses were conducted on different subgroups of target condition, in order to estimate the accuracy of the algorithm for each condition separately.”
“However, the main bottleneck of this type of research is the relative limited amount of data, which could possibly be overcome by systematic review and meta-analysis of the aggregated study results“ The systematic review is valuable, but I speculate that the main bottleneck cannot be overcome by the systematic review.
We revised the sentence as suggested by the reviewer; it now reads:
“However, one of the limitations of this type of research is the relative limited amount of data in each study, which could possibly be overcome by systematic review and meta-analysis of the aggregated study results [10].”
Ref. 10. “Therefore, as many previous review papers have pointed out, the existence of large data sets is indispensable for promoting ML study.” As shown in Ref. 10, I speculate that large dataset is more valuable than meta-analysis and/or systematic review.
We fully agree with the reviewer. This is now addressed at the end of the discussion section as an important recommendation for future research. It now reads:
“In addition, as MLAs are statistical model fitting methods, the reliability and performance of each study is at least partially depending on the study sample size. Therefore, a large dataset next to the available BraTS dataset would be indispensable for this field of research.”
“MLA-models needed to be at least internally validated.“ In several papers of Table 1, number of Validation set is NR. Are these papers valid for this review paper? If number of Validation set is not reported, the papers might be low-quality.
We agree with the reviewer that the lack of a validation set is a major drawback in scientific literature of MLAs in general. As shown by Van Leeuwen et al. (PMID: 33856519), various commercially available software packages were not reviewed in peer-review settings and external validation remains questionable as well. We elaborate on this limitation in the Discussion section of our manuscript where it reads:
4.1. Implementation of computer-aided approaches in future medicine
Diagnosis of different diseases by use of MLAs is believed to hold great potential in modern medicine. The number of the retrieved papers on this narrow topic is relatively limited, especially when compared to other reviews with a broader scope with regard to the use of artificial analysis in medical imaging analysis (61).61Nevertheless, conclusions and major limitations seem to be similar across fields. We can cautiously state that the accuracy of MLAs in non-invasive classification of glioma hold great potential and is equal to or better than the predictions of healthcare professionals. On the other hand, the lack of external validation of the obtained results was recognized as the major limitation of the current scientific literature. Additionally, poor reporting is known to be prevalent in MLA studies, which limits reliable interpretation of the reported diagnostic accuracy and thereby hampers clinical implementation. Improving reporting and publication will enable greater confidence in the results of future evaluations of these promising technologies in medicine. When such improved confidence will be achieved, prospective evaluation should be carried out in the context of an intended clinical pathway. With regard to the context discussed in this paper, MLAs could be used on the preoperative imaging data after which the predicted outcomes can be compared to the histopathological assessment after biopsy. Such implementation will help to elucidate whether important unknown covariates were present in the here reviewed retrospective studies. Thereafter, a randomized comparison could help to reveal and quantify possible clinical implications of implementing these MLAs in daily practice. Furthermore, as recently suggested by Bhandari et al., a greater effort is needed to start translating these findings into an interpretable format for clinical radiology [79]. In addition, as several studies focused on singular molecular biomarkers in gliomas, it must be underlined several molecular alterations in astrocytic, oligodendroglial gliomas can occur in different combinations [80]. Consequently, a growing number of (a combination of) different molecular tests are used to provide clinically relevant tissue-based biomarkers. Furthermore, the performance of MLAs to grade glioma according to the WHO grading score remains restricted by the interobserver variability of the neuropathological examination as reviewed by Van den Bent (2010) [81]. This clinically significant interobserver variation of the histological grading of glioma limits the diagnostic performance of other diagnostic tests (e.g., MLAs) as the neuropathological assessment was considered to be the ground truth. The aforementioned steps are pertinent to ensure that studies on the use of MLAs in non-invasive classification of glioma are of sufficient quality to evaluate the performance of the algorithms and will contribute to improved health care in the daily clinical practice.
In addition, a special section of the Discussion provides additional information on why we could not exclude these papers. Therefore, this should be regarded as a limitation of the current meta-analysis.
“Considering external validation as an additional inclusion criterium for a sub-analysis, we found that no meta-analysis could be performed. The twelve papers included in this review which validated their results externally investigated IDH mutation status (n=2), 1p/19q codeletion status (n=1), MGMT promotor methylation status (n=2), PTEN gene mutation (n=1), ATRX gene mutation (n=1), TERT gene mutation (n=1) and various predictions with regard to WHO grading (n=4), indicating that no pooled data can be acquired from these individual studies. Although computer-aided classification of glioma holds great potential, computer-obtained diagnosis is not likely to replace histopathologic diagnosis in the near future.”
“Five of the included studies described multiple target conditions and/or multiple tested MLA-models, each of those presented separately in Table 2.” Where is Table 2?
We sincerely apologize for this inconvenience as Table 2 was indeed lacking of our submitted manuscript. However, in the revised manuscript we added the information of Table 2 to the information of Table 1 and removed Table 2 from the revised manuscript. The information can now be appreciated in Table 1. Consequently, the text was revised as well to ensure proper referencing.
Minor points
“Papers describing the use of MLAs for the classification of molecular characteristics of gliomas were reviewed.” Please clarify that authors included the paper where binary classification of molecular characteristics of gliomas was evaluated using MLAs.
We clarified that we included papers in which binary classification of molecular characteristics of glioma were evaluated by use of MLAs as requested by the reviewer. Please see the revised text.
“. [14]” typo.
Typographical error was corrected.
“For 12 of the included studies an external out-of-sample validation was carried out” Please summarize the performance in external validation as Table for the 12 studies. Please include number of external validation set in the Table.
Additional information on the external validation set is provided in the revised version of Table 1.
Table 1. If cross validation is used, how are numbers of Training set and Validation set reported?
We reported the results of the Cross Validation set. This is now clarified in the Legend of Table 1.
Table 1. If dataset is divided into training/validation/test sets, how are numbers of these sets reported?
We reported the results of the validation sets. This is now clarified in the Legend of Table 1.
Abbreviation should be added to Table 1. For example, “T1w+c”, “NR”, “LGG/HGG”
A legend was added to Table 1 as requested by the reviewer.
Why is 95% CI of AUC higher than 1.0 in Figures?
The reviewer is correct; analyses were re-run with AUC values being restricted to 0.0-1.0 as is correct. Please see the revised and redesigned Figures.
Round 2
Reviewer 1 Report
Authors responded my comments very well. I do not have any further question. Congratulations!
Author Response
We would like to thank the reviewer for their detailed comments and suggestions for the manuscript.
Reviewer 2 Report
- "When the dataset was split into training/validation/test sets, the Performance values of the Validation set were provided here." When the dataset was split into training/validation/test sets, usage of validation/test sets might be different between papers. One paper might use validation set for hyperparameter tuning, and another paper might use it for performance evaluation. In Table 1, authors must provide classification metrics in the set used for performance evaluation.
- If Figure 3 of original submission was removed, Figure of the funnel plot (Figure 7 of revision)
Author Response
Thank you again for your insightfull comments. We have double-checked all metrics of the papers in which the dataset was split into training/validation/test sets in order to assure that the provide classification metrics are indeed those which were reported for performance evaluation of the MLAs. Please see Table 1 in the revised manuscript. We found that, indeed, only classification metrics which displayed the performance evaluation results were presented. No other metrics (e.g. metrics of the hyperparameter tuning) were presented in Table 1. This is now also mentioned more clearly in the Table caption where it now reads:
"If cross validation was used, the Performance values of the cross validation set were provided here. When the dataset was split into training/validation/test sets, the Performance evaluation values with regard to the investigated outcome (e.g., IDH genotype) of the Validation set were provided here."
As requested, Figure 7 (Funnel plot) was also removed from the manuscript.